# Tongue Posture, Tongue Movements, Swallowing, and Cerebral Areas Activation: A Functional Magnetic Resonance Imaging Study

**Fabio Scoppa [1,2], Sabina Saccomanno [3,*], Gianluca Bianco [4] and Alessio Pirino [5]**

[1]  Master's Degree Course in Posturology, Faculty of Medicine and Dental Surgery, University of Rome "Sapienza", 00185 Rome, Italy; fabioscoppa@chinesis.org

[2]  Chinesis I.F.O.P. Istituto di Formazione in Osteopatia e in Posturologia, 00152 Rome, Italy

[3]  Orthodontic Residency, Department of Life, Health and Environmental Sciences, University of L'Aquila, 67100 L'Aquila, Italy

[4]  Research Laboratory of Posturology and Neuromodulation RELPON, Master's Degree Course in Posturology, Faculty of Medicine and Dental Surgery, University of Rome "Sapienza", 00185 Rome, Italy; gianlucabianco@libero.it

[5]  Department of Biomedical Sciences, University of Sassari, 07100 Sassari, Italy; axelpir@uniss.it

*  Correspondence: sabinasaccomanno@hotmail.it; Tel.: +39-339-4153290

**Abstract:** The aim of this study was to pinpoint the cerebral regions implicated during swallowing by comparing the brain activation areas associated with two different volitional movements: tongue protrusion and tongue elevation. Twenty-four healthy subjects (11—males 22 ± 2.9 y; 13—females 23 ± 4.1 y; were examined through functional magnetic resonance imaging (fMRI) while performing two different swallowing tasks: with tongue protrusion and with tongue elevation. The study was carried out with the help of fMRI imaging which assesses brain signals caused by changes in neuronal activity in response to sensory, motor or cognitive tasks. The precentral gyrus and the cerebellum were activated during both swallowing tasks while the postcentral gyrus, thalamus, and superior parietal lobule could be identified as large activation foci only during the protrusion task. During protrusion tasks, increased activations were also seen in the left-middle and medial frontal gyrus, right thalamus, inferior parietal lobule, and the superior temporal gyrus (15,592-voxels; Z-score 5.49 ± 0.90). Tongue elevation activated a large volume of cortex portions within the left sub-gyral cortex and minor activations in both right and left inferior parietal lobules, right postcentral gyrus, lentiform nucleus, subcortical structures, the anterior cingulate, and left insular cortex (3601-voxels; Z-score 5.23 ± 0.52). However, the overall activation during swallowing tasks with tongue elevation, was significantly less than swallowing tasks with tongue protrusion. These results suggest that tongue protrusion (on inferior incisors) during swallowing activates a widely distributed network of cortical and subcortical areas than tongue elevation (on incisor papilla), suggesting a less economic and physiologically more complex movement. These neuromuscular patterns of the tongue confirm the different purpose of elevation and protrusion during swallowing and might help professionals manage malocclusions and orofacial myofunctional disorders.

**Keywords:** brain; tongue posture; tongue elevation; tongue protrusion; fMRI analysis; swallowing

---

## 1. Introduction

Swallowing is a complex sensorimotor function involving volitional and reflexive activities to move saliva, liquids, and solid food from the oral cavity, through the pharynx and into the esophagus [1–4]. Current knowledge of the neuro pathophysiology of the tongue during swallowing events is slowly

advancing. This is partially related to the ever increasing knowledge regarding specific tongue activities and due to the discovery of five kinds of exteroceptors on the palate that connect to the nasopalatine nerve [5,6]. In physiological conditions, the tongue during deglutition touches the palate behind the incisal papilla and its tip presses upon this point. This area is also called "spot" by myofunctional therapists and seems to act as a natural relay between the palate receptors and the cerebral cortex [7,8]. It has been found that in patients suffering from dysfunctional swallowing, the tongue in resting conditions may be placed far from the palate, due to the prevalent activity of the genioglossus muscle, characteristic of a tongue often thrusting forward or between the dental arches [9,10]. The genioglossus muscle is the largest in the tongue followed by the transverse, together representing almost 40% the tongue's mass (Stone et al., 2018) [11]. Since both sets of muscles are involved in keeping the airways patent by displacing the tongue anteriorly (tongue thrust), the expectation is that they will cause different activations of cortical and subcortical areas than during swallow with tongue thrust than tongue against the palate. In clinical diagnosis, patients who do not keep the tip of the tongue on the "spot" during swallowing, frequently present postural abnormalities, and an incorrect distribution of the body weight [9]. The functional magnetic resonance imaging (fMRI) technique, has been used for over twenty years to investigate cortical representations during swallowing activities [12–19]. Swallowing has been suggested to be multifocal [20–23], being associated with the activation of insular regions of the cortex, the premotor/sensorimotor cortices and the anterior cingular gyrus [18,24–27]. Cortical activation has also been reported to be associated to volitional swallowing of saliva and water bolus [28,29]. The most prominent swallow-related activation foci correspond to the lateral pericentral and perisylvian cortices, the anterior cingulate cortex (ACC) and adjacent supplementary motor area (SMA), the right insula, operculum, and precuneus. Moreover, the tongue elevation activates a larger total volume of cortex compared to swallowing activities with significantly greater activation within the ACC, SMA, right precentral and postcentral gyri, premotor cortex, right putamen, and the thalamus [30], caused by the stimulation of the palatal spot. In addition, a number of studies have reported lateralization of sensorimotor cortical activation [21,22] whereas others have identified more bilateral activations [23–33], with handedness-independent hemispheric dominance [14,22,34]. We think our study complemented most of the other ones mentioned in this article, as each one offers a different view of something inherently complex such as the neuronal interplay between cognition and the sensory–motor behavior of the tongue, at least in two very specific movements.

It has been documented for decades that there are two types of swallowing movements and that the one with tongue protrusion is not physiological because it is accomplished with a significant recruitment of other muscles. What was missing was a clear neurophysiological counterpart that this study aimed to solve by visualizing the additional recruitment of neurological areas, to confirm that a swallow with tongue protrusion is not efficient from a neuromuscular point of view. In the future, other studies may focus on the tongue protrusion swallow typical of people with insufficient nasal breathing or those with sleep apnea and all the cases that require the study of the functions swallowing and breathing with implications of the cerebral response. From the analysis of the literature it emerges that the studies done previously to date have dealt with the evaluation of swallowing to correlate damaged areas of the brain with the presence or type of dysphagia. Many studies focus on the effects of stroke on swallowing, while fewer clinical studies have focused on swallowing in other neurological conditions such as Parkinson's disease, Alzheimer's disease, or traumatic brain injury. Regarding the central control of atypical and physiological swallowing, clinical studies are lacking, and those available are dated. Therefore, the aim of this investigation was to elucidate the cortical representation of two swallow-related motor tasks, namely tongue protrusion (simulated dysfunctional swallowing) and tongue elevation (physiological swallowing) through the use of fMRI.

The hypothesis was that tongue protrusions is a compensatory movement and not a physiological occurrence, and therefore it may activate more cortical and subcortical areas related to movement coordination, as opposed to tongue elevation, which is present in physiologic swallow, and therefore it may activate more sensorimotor cortical areas rather than subcortical areas.

## 2. Materials and Methods

### 2.1. Study Design, Setting and Subjects

The STROBE statement was adopted [35–37]. A case-control study was carried out, with participants who become, afterwards, their own controls. The two experimental conditions (Elevation; Protrusion) were balanced with a randomized controlled order. The study was carried out at the Department of Neurology and Otorhinolaryngology, Sapienza University, Rome, Italy. Twenty-four healthy subjects (11 Males 22 ± 2.9 y; 173 ± 5.2 cm; 79 ± 8.9 kg; and 13 Females 23 ± 4,1 y; 159 ± 4.4 cm; 61 ± 6.7 kg) were recruited through a proper advertisement within the University Campus. A pre-screening through health history was obtained from 40 potential participants of which 24 were enrolled. All those excluded presented with (1) a history of gnathologic and odontostomatological disorders, (2) tongue pathologies, (3) neurologic disorders, and (4) symptoms of odontostomatological pathology, (5) a short lingual frenum [38], or (6) periodontal problems [38–42].

All subjects recruited were right-handed and all were informed of the examination procedure and the commitment required. The ethical principles, regarding human experimentation, were adopted according with the Declaration of Helsinki by the World Medical Association (WMA). The study protocol was approved by the University of Rome "Sapienza" Department of Neurology and Otorhinolaryngology and by the Ethics Committee of the University of L'Aquila, Italy (Document DR206/2013, 16 July 2013).

The data analysis was performed retroactively, as a retrospective study [43,44], and approved by the Scientific Commission of the Department of Biomedical Sciences, University of Sassari, Italy.

### 2.2. Functional Imaging Acquisition

Measurements were done using a Siemens 1.5 Tesla whole body magnetic system (Siemens Medical Systems, Erlangen, Germany) with echo-planar capabilities and a head volume radio-frequency coil, equipment of DEA University of Rome "Sapienza". The blood oxygenation level dependent (BOLD) functional images of the whole brain were acquired as magnetic susceptibility (T*2)-weighted echo-planar images (64 × 64 matrix size, 240 mm field of view) using the following parameters: repetition time TR = 3000 ms, echo time TE = 50 ms, flip angle $\alpha = 90°$, mono excitation. Functional data were collected from 36 contiguous, 4 mm-thick axial slices oriented parallel to the bi-commissural plane, then acquiring volumetric T2-weighted images (256 × 256 matrix size, 240 mm field of view, 192 slices). [17].

### 2.3. fMRI Standard Procedure

All 24 subjects participated in an 18-min total time functional imaging run during a single experimental session, while laying supine. The functional run was composed of 2 different "cortical" activation tasks, each lasting 9 min, and these were performed in a controlled order: first 12 participants with (1) Tongue Protrusion; (2) Tongue Elevation and then 12 participants with (1) Tongue Elevation; (2) Tongue Protrusion. Each task had a duration of 30 s, similar to the timing protocol by Kern et al. (2001) [45]. The rationale for each task being 30 s is that a functional resonance of a just a few seconds on 4 mm thick slices would not be enough to have accurate information [15,16].

Moreover, the subjects were instructed to swallow accumulated saliva, which requires some time to naturally be produced and although healthy subjects could swallow between 3 and 7 times in 30 s (Affoo et al. 2015), [46]. We felt that not imposing the additional strain of producing multiple saliva swallows would give us a better reading. A physiological swallow lasts between 1 s and 1.5 s, depending on many factors, including the viscosity of saliva or food [2,3,47] and Martin–Harris, 2008; [48] Logemann JA [49].

Therefore, the rationale for longer preparation times (compared to the time needed to swallow a few seconds) is linked to the type of investigation and refers to the time required from each patient to obtain the recording of three-dimensional images acquired at a rate every 2 or 3 s. To be sure of having

more precise information it is necessary to acquire hundreds of images and for this reason we settled on 30 s intervals.

For brain structures, the thickness of 4 mm instead of 2 mm is the right compromise to have a greater number of point of data acquisitions, reduced noise, and therefore a reduction of the artifacts and an increase in the image quality. Furthermore, taking a greater thickness during functional resonance test generates a greater number of ossiHb which contributes to the signal from each layer further improving the quality of the exam. Taking 2 mm slices would have reduced these outcomes, hence the choice of 4 mm.

For brain structures, the thickness of 4 mm was deemed to be the appropriate choice to have a greater number of point of data acquisitions, reduced noise, and therefore a reduction of the artifacts and an increase in the image quality. Furthermore, taking a greater thickness during functional resonance test generates a greater number of ossiHb, which contributes to the signal from each layer further improving the quality of the exam. Taking 2 mm slices would have reduced these outcomes, hence the choice of 4 mm, chosen by the radiologist who performed this examination based on his professional experience. Furthermore, a study on functional magnetic resonance confirms that unlike magnetic resonance, the thickness of 4 mm is recommended [14,49,50].

Task switching was indicated using acoustic cues at the beginning and end of each block of tasks or rest. Participants were instructed just before each task, in order to correctly perform the tasks. The swallowing had to be performed only once for each task, without exaggerating any oral movements. (Figure 1).

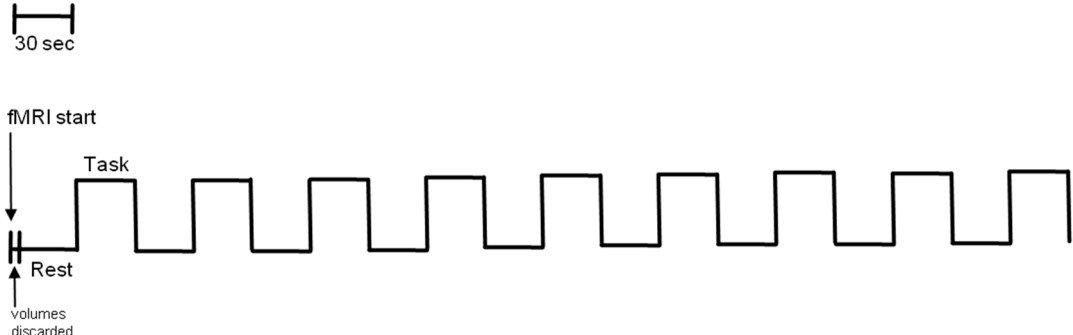

**Figure 1.** Study design. Nine minutes of functional magnetic resonance imaging (fMRI) in which participants performed only protrusion and afterwards 9 min of only elevation characterized the experimental condition. The order was controlled as follow (12 subjects with protrusion and elevation; 12 subjects with elevations and protrusion).

The issue arose that the first task sequence could influence the cerebral effect of the second, but we could not find publications on the subject to offer a reasonable guidance. Moreover, it was considered that in order to have a sufficiently long wash-out period (time needed for the cerebral effect of the first sequence to wear off), we would have needed to perform two fMRI sessions, triggering concerts about its ethical justification and economic sustainability. Therefore, we considered our choice as guided by clinical experience, since we were lacking peer-reviewed guidance. Our choice could guide future studies in considering more appropriate ways to eliminate unwanted side effects that could bias the outcomes.

The standardized procedure for the tongue elevation swallowing task was to raise the body of the tongue to the palate touching the "spot" and then swallow the saliva. This was considered as a physiological swallowing. The standardized procedure for the tongue protrusion task was to place the tongue tip against the lingual surface of the lower incisor. This was considered a dysfunctional swallowing.

*2.4. fMRI Data Analysis*

Statistics and images registration were performed with the Statistical Parametric Mapping (SPM 8) software written by the Wellcome Department of Cognitive Neurology, University College, London, UK) [51]. For each section, the three volumes were discarded according to tissue relaxation artifacts. Data preprocessing included realignment of the images with the first image of the series in order to adjust for motion artifact and stereotactic normalization. Such alignment followed the Montreal Neurological Institute (MNI) coordinates approximation of Talairach et al. [43,52], and were then smoothed with a Gaussian spatial filter to a final smoothness of 8 mm. Brain dimensions were then scaled to the size of the extremes of the cerebellum in the direction of the defined axe. The effect of the experimental paradigm was estimated on a voxel-by-voxel basis. Data from each voxel was modeled using the general linear model with separate hemodynamic response functions and their time derivatives modeling each period of each task [53,54]. The data points were involved with the hemodynamic response function chosen to represent the relationship between neuronal activation and blood flow changes. Within each region of statistical significance, local maximum of signal increase was determined (the voxels of maximum significance) and their location expressed in terms of MNI x, y, z coordinates. At a later time, through the STATISTICA software version 8.0 the variable (Cluster Size) was treated as independent samples (Protrusion vs. Elevation) and categorized by Common Activated Areas (CAA) and Uncommon Activated Areas (UAA). A cluster is a measure of the difference between groups obtained from maps of voxel, considering as well statistical tests and the spatial cluster, and it indicates a specific brain area, while a cluster size is the sum of the voxels of all patients activated in a specific brain area (cluster). Therefore, to check the error we constructed the distribution of the maximum cluster for each statistical map and the largest cluster was considered.

It was decided that considering clusters of spatially close voxels was the best way to analyze our fMRI data of the activation areas, mostly because cluster size was considered a variable per se and analyzed based on the two independent samples of elevation and protrusion. We kept and listed the voxel size to further describe the cortical activation. It was decided that for the purpose of our study, we would choose cluster-wise significance of 0.01 uncorrected for the images and corrected for the two tables describing the various areas activated. Although there are added benefit to the speed of processing time and statistical analysis, the use of a "mask" analysis was not considered at that time.

## 3. Results

*fMRI*

Multiple areas of increased cerebral activation were observed in both tongue tasks during swallowing. (Appendix A)

The precentral gyrus and the cerebellum were activated during both swallowing tasks while the postcentral gyrus, thalamus, and superior parietal lobule could be identified as large activation foci only during the tongue protrusion task. During the protrusion task minor activations were also seen in the left-middle and medial frontal gyrus, right thalamus, inferior parietal lobule, and the superior temporal gyrus (15,592-voxels; Z-score 5.49 ± 0.90). The Right Superior Temporal Gyrus was activated by auditory stimuli and reported in Table 1. Given the same acoustic signal that preceded each task, its activation was expected, and although it was mentioned in the results text and in Table 1, it was not considered in the analysis of the swallowing task, although it was interesting that a minor RSTG activation would be present only during tongue protrusion. It would be interesting in a future study to see how a visual cue, rather than an auditory one, would affect both or either swallowing task. From the study by Katie J Lasota et al., 2003, it emerges that even if it is demonstrated that there is a dependence of the activation of the primary auditory cortex on the level of stimulus intensity; however, the coding of intensity at the level of the auditory cortex may not being a linear process moreover it is the intensity of the sound that can influence the occurrence and the bilateral nature of the auditory cortical activity. All of this affects the studies with functional magnetic resonance; therefore, the authors

recommend further studies to clarify the implications of functional magnetic resonance on the auditory cortex [50].

**Table 1.** Significant activation areas during protrusion swallowing tasks.

| Cluster | Cluster Size | Brodman Area | MNI Coordinates | | | Z Score | Structure |
|---|---|---|---|---|---|---|---|
| | | | x | y | z | | |
| 1 | 7743 | 1, 6 | 48 | −8 | 34 | 7.30 | R. PrecentralGyrus |
| 2 | 6165 | 1, 4, 6 | −56 | −8 | 34 | 7.00 | L. PrecentralGyrus |
| 3 | 448 | | 18 | −60 | −22 | 6.11 | R. Cerebellum |
| 4 | 407 | | −16 | −62 | −18 | 5.54 | L. Cerebellum |
| 5 | 275 | 3 | −16 | −28 | 62 | 5.49 | L. PostcentralGyrus |
| 6 | 200 | 7 | 36 | −60 | 64 | 5.38 | R. Superior Perietal Lobule |
| 7 | 232 | | −10 | −14 | 6 | 5.35 | L.Thalamus |
| 8 | 33 | 3, 6 | −34 | 8 | 40 | 5.33 | L. Middle Frontal Gyrus |
| 9 | 24 | | 12 | −14 | 0 | 4.69 | R. Thalamus |
| 10 | 29 | 1, 6 | −8 | −4 | 64 | 4.66 | L. Medial Frontal Gyrus |
| 11 | 17 | 39 | 56 | −56 | 12 | 4.60 | R. Superior Temporal Gyrus |
| 12 | 19 | 40 | 56 | −52 | 52 | 4.50 | R. Inferior Parietal Lobule |

Coordinates x, y, z represent the position of the voxel with peak activation level (One-sample *t*-test, uncorrected $p < 0.001$, corrected at the cluster level) with cluster (in mm) relative to the anterior commissure (AC) in the stereo-tactic space; x:Lateral distance from the midline (+right, −left); y: Anteroposterior distance from the AC (+anterior, −posterior); and z: height relative to the AC line (+higher, −lower).

Tongue elevation activated a large volume of cortex portions within the left sub-gyral cortex and minor activations in both right and left inferior parietal lobules, right postcentral gyrus, lentiform nucleus, sub-gyral lobe, the anterior cingulate, and the left insular cortex (3601-voxels; Z-score 5.23 ± 0.52). The regions which demonstrated larger area activation during the tongue protrusion in swallowing are shown in detail in Figure 2 and Table 1. The total brain volume activated in this case was 15,592 voxels and the spatial activation patterns within the left and right hemispheres were clearly distinct. Stronger activations (largest voxel clusters) were observed in the postcentral gyrus, in both left (i.e., Brain Area BA 1, 4, 6) and right (i.e., BA 1, 6) hemispheres (bilateral activation), with a modest predominance in the right hemisphere. A second prominent area of activation corresponded, bilaterally, to the cerebellum (the size of the activation focus within the left hemisphere was substantially larger, suggesting a moderate left hemisphere groupwise functional lateralization), the left postcentral gyrus (i.e., BA 3), and the right superior parietal lobule (i.e., BA 7). The cortical activation was also identified within several subcortical nuclei, including the thalamus bilaterally (1.6% of the total number of significant voxels), with the activation volume in the left hemisphere being 10 times greater than the volume in the right hemisphere. Less consistent activations were seen in left-middle (i.e., BA 3, 6) and medial-frontal gyrus (i.e., BA 1, 6), right inferior parietal lobule (i.e., BA 40), and right superior temporal gyrus (i.e., BA 39). These areas where a minor activation was showed represented only 0.6% of the total number of significant voxels.

The clusters of cortex activated during the tongue elevation swallowing task are shown in Figure 3 and Table 2. The total activated volume of the brain during the elevation-task was 3601voxels. The largest activation pattern was observed in the left precentral gyrus (i.e., BA 1, 4, 6) bilaterally, with a size of the activation focus within the left hemisphere substantially larger than the contralateral, suggesting a left hemisphere functional lateralization for tongue-elevation during swallowing. Other significant activation clusters included left inferior and parietal lobule and the cerebellum bilaterally (left size greater than the right one). Smaller areas of activation corresponded to the inferior parietal lobule bilaterally, right postcentral gyrus, lentiform nucleus, infer parietal lobule, and anterior cingulate and left insula. These smaller areas represented only 7.7% of the total number of activated voxels. Interestingly, while watching the data with CAA (Protrusion mean 2956.4 ± 3695.5 vs. Elevation mean 616.0 ± 688.2; *t*-test 0.20) vs. UAA (Protrusion mean 57.85 ± 98.2 vs. Elevation mean 37.2 ± 70.6; *t*-test 0.52), we found a heterogeneity of variance in cluster sizes.

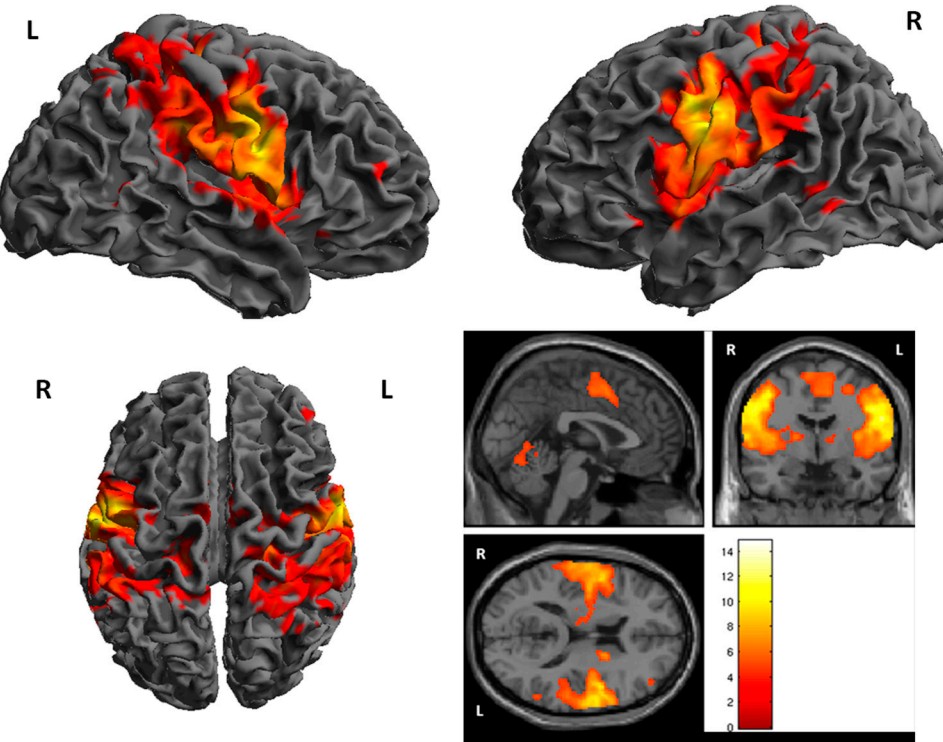

**Figure 2.** Cortical activation observed during the tongue protrusion swallowing. Clusters of significant activation (*p* < 0.0001, uncorrected) are displayed on normalized sagittal (up-left panel), coronal (up-right panel), and axial (down panel) brain slices. Color scale indicates z score range.

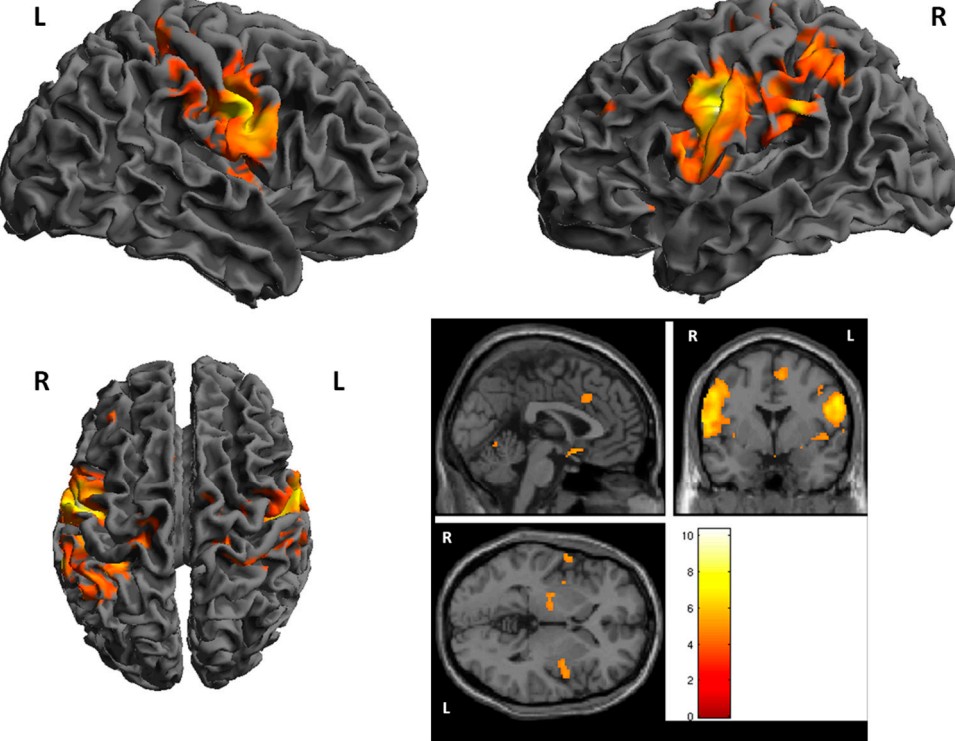

**Figure 3.** Cortical activation observed during the tongue elevation swallowing. Clusters of significant activation (*p* < 0.0001, uncorrected) are displayed on normalized sagittal (up-left panel), coronal (up-right panel), and axial (down panel) brain slices. Color scale indicates z score range.

**Table 2.** Significant activation areas during elevation swallowing tasks.

| Cluster | Cluster Size | Brodman Area | MNI Coordinates | | | Z Score | Structure |
|---|---|---|---|---|---|---|---|
| | | | x | y | z | | |
| 1 | 1637 | 1, 6 | −46 | −10 | 36 | 6.25 | L. PrecentralGyrus |
| 2 | 231 | | −16 | −60 | −22 | 5.67 | L. Cerebellum |
| 3 | 180 | | 26 | −62 | −22 | 5.56 | R. Cerebellum |
| 4 | 1013 | 1, 6 | 52 | −8 | 26 | 5.43 | R. PrecentralGyrus |
| 5 | 86 | 40 | −64 | −38 | 30 | 5.42 | L. Inferior Parietal Lobule |
| 6 | 62 | | 22 | −4 | −10 | 5.31 | R. Lentiform Nucleus |
| 7 | 261 | 3, 40 | −34 | −40 | 42 | 5.20 | L. Inferior Parietal Lobule |
| 8 | 64 | 5 | 46 | −38 | 64 | 5.00 | R. PostcentralGyrus |
| 9 | 19 | 40 | 54 | −26 | 30 | 4.55 | R. InferiorParietalLobule |
| 10 | 14 | 1, 25 | 4 | 6 | −14 | 4.50 | R. AnteriorCingulate |
| 11 | 16 | 13 | −48 | 10 | −8 | 4.50 | L. Insula |
| 12 | 18 | 40 | 30 | −40 | 62 | 5.45 | R. InferiorParietal Lobule |

Coordinates x, y, z represent the position of the voxel with peak activation level (One-sample *t*-test, uncorrected $p < 0.001$, corrected at the cluster level) with cluster (in mm) relative to the anterior commissure (AC) in the stereo-tactic space; x:Lateral distance from the midline (+right, −left); y: Anteroposterior distance from the AC (+anterior, −posterior); and z: height relative to the AC line (+higher, −lower).

## 4. Discussion

The present investigation suggests that both tongue elevation and protrusion during swallowing activate a widely distributed network of cortical and subcortical areas, but with different configurations. The most consistent finding was recruitment of the bilateral primary motor and premotor cortex, bilateral supplementary motor area (SMA) [55] and the cerebellar activation during these tasks. A possible explanation could be that swallowing tasks involve cooperation of the motor components in planning tongue protrusion and elevation and refining oropharyngeal movements based upon sensory input by the cerebellum. The activation of these areas during oropharyngeal movements such as speech production [56], perception [57], and swallowing have been reported by a consistent number of studies [24,31,33]. According to these studies [18,24,33], the primary motor cortex and the cerebellum are activated only during repetitive saliva swallowing tasks, consistent with our investigation on tongue protrusion and elevation, and there is at least one study that shows that the repetitive saliva swallowing increases demands on the "swallowing" neuronal network [33,34]. In contrast, other studies that did not consider repetitive saliva swallowing [19–37,43,51–58], did not observe the foci activation. There is emerging evidence showing how the premotor areas are not solely involved in motor programming and movement initiation but also in sensory processing [59] and may be regarded as supplementary sensorimotor areas [60]. In our investigation, despite similar activations between tasks of tongue protrusion and elevation in swallowing, the cluster size of brain activation for the protrusion task was larger than the elevation task and differently distributed. This may have occurred because protrusion involved greater motor effort than elevation, suggesting that swallowing with the tongue in protrusion requires the involvement of many regions in the cerebral cortex and subcortex. Moreover, due to the specific physiology of the genioglossus muscle, [61]. which is the main muscle involved in maintaining the airways patent, a tongue trusting is a well-documented physiological response to airways disturbances, and it involves compensatory sensory motor recruitment reflected in the increased cortical and subcortical activation.

Consistent with the principles of neuroplasticity employed during swallowing therapy or in myofunctional therapy, practicing a task changes the activation patterns within the cortical network involved in each task or produces functional reorganization, which indicates that the brain regions recruited for each task are altered as a function of practice [62,63]. It is also well recognized that a highly practiced task requires less brain activation when compared to a less practiced one or a task's initial learning phase [64]. Further practice may eventually lead to new changes in the cortical motor outputs. This means there is a pattern of decreased global activation as well as a shift in activity from cortical to subcortical areas as automaticity is achieved [65,66], important for generalization and habituation of

correct behavior during therapy. These findings support the hypothesis that the cerebellum is primarily involved in the early phase of motor sequence learning, as well as the basal ganglia, that can possibly contribute to an automatization phase [67,68]. Since many precentral areas are recruited during tongue protrusion in deglutition, it may be posited that such a task involves neuromuscular compensations and therefore it can be considered part of a dysfunctional swallowing pattern.

The activated areas we found lateralized to the left hemisphere during tongue protrusion, including part of the somatosensory cortex (postcentral gyrus), were less evident in the right lateralized activation during tongue elevation. Interestingly, there is evidence suggesting that the left postcentral cortex is specialized for oral sensorimotor functions [31].

We found that a number of cortical regions were activated only by tongue protrusion or by tongue elevation. Activation of the somatosensory gyrus, the posterior parietal cortex, the superior lobule and the inferior lobule, are related to attention, selection of stimuli, sensorimotor functions and integration, with different activation on the right and on the left hemisphere, as seen in the images 2 and 3. These activations have been confirmed in other studies [27,28,31,69,70] and suggest that the volitional nature of the tongue protrusion in this study, compared to the tongue elevation, combined with a larger compensatory recruitment of muscles activated in protrusion, is manifested in the activation of a larger network of neurons. The nature of the task with an auditory signal to change activity could have contributed to the activation of the posterior attention system of the parietal lobule.

These findings have clinical application as they confirm the economy of brain activation, both cortical and subcortical, in the physiological tongue-palate interaction, which is promoted during oral myofunctional therapy and dysphagia therapy. This study suggests as well that the nature of tongue protrusion, being a common occurrence in restricted airways (presence of hypertrophic tonsillar tissue, sleep disordered breathing, or genetic disorders), recruits and organizes other neuromuscular segments resulting in enhanced activation of various cortical and subcortical areas, as confirmed by previous studies [23,27,28,55,69,71–74] and as described in the table below.

Our results seem to suggest that activation of the anterior cingulate during tongue elevation in swallowing may support the involvement of this area in swallow-related motor tasks as a non-specific site for attention associated with other motor tasks in addition to deglutition which are part of the compensatory sensory–motor pattern during a tongue protrusion posture or swallow. In addition, the anterior cingulate is a recognized center for the processing of painful stimuli and of stimuli linked to emotions [14,23,34,75–77].

In the present investigation, tongue elevation activated a number of brain areas that were not active during tongue protrusion, within the right lentiform nucleus (putamen and globus pallidus), the anterior cingulate, and the insula. However, no thalamus activation was observed during tongue elevation. Conversely, an activation within the thalamus and supplementary motor area during tasks of tongue protrusion in swallowing is consistent with saliva swallowing being mediated by this thalamo-cortical motor circuit [28,73,74] as the thalamus, and especially the ventral lateral nucleus, may represent a relay for basal ganglia outflow to the supplementary motor area. This is further confirmation, consistent with results of other studies mentioned, that the economy of sensorimotor activation of the tongue in elevation, compared to the greater cortical and subcortical sensorimotor activation in protrusion, is a guiding principle in promoting tongue elevation in patients during therapy as this position or movement is less complicated and less dependent on cognitive skills, in absence of overriding breathing issues that change the overall pattern of tongue posture at rest and during swallow., [44]. Bocquet et al., [78] suggest we might suppose a reduction of motor unit recruitment during tongue elevation task, suggesting a reduction in energy expenditure.

According to our results and recent literature, the swallowing patterns might help professionals to manage malocclusions and related disorders. Dental malocclusion and the oral muscles dysfunction have been previously investigated in patients with atypical swallowing by Saccomanno et al. [38,79,80]. The authors concluded that the optimal approach must include also myofunctional treatment in order to re-establish physiological tongue position and swallowing. Bocquet et al. in 2008 [78] already

examined whether dysfunctional swallowing influences postural parameters as well. Those authors found that swallowing would have the same postural effects as cognitive tasks by increasing the postural oscillations and the energy spent by the postural system [78].

We consider this study a pilot intervention with several concerns and limitations, but with a promising take-home message for all those professionals who are dealing everyday with malocclusions and related disorders [79]. In addition, this study seems to support the clinical usefulness of strategies and exercises promoting tongue rest position against the palate and retraining of swallowing with the upward movement of the tongue against the palate [80].

In retrospective this study should have analyzed the between-group effect to verify if indeed the randomized assignment of the two tasks had a significant effect and whether there were any differences. Instead we chose to aggregate the data and focus only on the cumulative differences between the neurological representation of the two tasks.

Other limitations to this study that we hope are addressed in the future are better verification of the cross-over design to eliminate unwanted side effects of the study, especially since we considered a wash-out period as not feasible; the use of other forms of triggers for tasks, rather than a sound, or including in the overall brain function analysis the impact of the sound on the task itself; and the use of a mask to better analyze the fMRI data.

## 5. Conclusions

The results of our study suggested that tongue protrusion (on inferior incisors) during swallowing activates a more widely distributed network of both cortical and subcortical areas than tongue elevation (on incisor papilla), suggesting a less economic and a physiologically more complex movement. Conversely, although tongue elevation activated several areas related to sensorimotor integration, there was an evident "economy" in activation, even in a volitional task, suggesting a more physiological swallowing pattern with less involvement of subcortical structures necessary to coordinate multiple functions. These neuromuscular patterns of the tongue confirm the different purpose of elevation and protrusion during swallowing and assist professionals in evaluate and manage them in presence of malocclusions and orofacial myofunctional disorders.

The fMRI is to date the most a valid diagnostic test, which also seems to validate the results of swallowing studies done by William Proffit in the 1970s on Australian Aborigines regarding the vertical vector of the tongue pressure against the palate [81] The so called "anterior tongue thrust" has been known to be related to malocclusion, as the vector of force is exerted forward as opposed to upwards (elevation). However, it is clinically important to detect the "tongue thrust" that happens for lack of optimal nasal breathing and during sleep to prevent occlusion of the airways, but that occupies different neural pathways, as indicated in our study and in others.

The present study is clinically relevant as it connects the results of the fMRI with the principles of neuroplasticity involved in the practice of various tongue positions during swallowing and a guiding principle in the application of myofunctional therapy and other therapies in general.

Finally, this study suggests a sensorimotor role of the pressure of the tongue which is a fundamental drive on bone metabolism, growth, and development, constantly changing based on age and gender, as already reported by Yuko Fujita [82]. These aspects, therefore, could represent the start of further studies.

**Author Contributions:** All authors contributed equally on this paper. All authors have read and agreed to the published version of the manuscript.

**Funding:** This research received no external funding.

**Acknowledgments:** We are grateful to all contributors for their patience. Thanks to Licia Coceani Paskay, MS, CCC-SLP, Los Angeles, CA, USA, who provided expert advices and contribution.

**Conflicts of Interest:** The authors declare that they have no competing interests.

## Appendix A

**Table A1. Description some brain areas**.

| | |
|---|---|
| **Precentral gyrus** | The precentral gyrus is responsible of the control of voluntary movement. |
| **Postcentral gyrus** | It is the location of the primary somatosensory cortex, the main sensory receptive area for the sense of touch. |
| **Superior parietal lobule** | Involved with spatial orientation. |
| **Inferior parietal lobule** | Involved in the perception of emotions in facial stimuli, and interpretation of sensory information. Moreover, is concerned with language, mathematical operations, and body image, particularly the supramarginal gyrus and the angular gyrus. |
| **Medial frontal gyrus** | The right middle fontal gyrus (MFG) has been proposed to be a site of convergence of the dorsal and ventral attention networks, by serving as a circuit-breaker to interrupt ongoing endogenous attentional processes in the dorsal network and reorient attention to an exogenous stimulus. |
| **Lentiform nucleus** | Part of the basal ganglia. The basal ganglia are associated with a variety of functions, including control of voluntary motor movements, procedural learning, habit learning, eye movements, cognition, and emotion. |
| **Anterior cingulate** | The superior temporal gyrus contains the auditory cortex, which is responsible for processing sounds. Specific sound frequencies map precisely onto the auditory cortex. This auditory (or tonotopic) map is similar to the homunculus map of the primary motor cortex. |
| **Superior temporal gyrus** | The superior temporal gyrus contains the auditory cortex, which is responsible for processing sounds. Specific sound frequencies map precisely onto the auditory cortex. This auditory (or tonotopic) map is similar to the homunculus map of the primary motor cortex. |
| **Insular cortex** | The insula controls autonomic functions through the regulation of the sympathetic and parasympathetic systems. It has a role in regulating the immune system. |
| **Thalamus** | The thalamus relays sensory impulses from receptors in various parts of the body to the cerebral cortex. A sensory impulse travels from the body surface towards the thalamus, which receives it as a sensation. This sensation is then passed onto the cerebral cortex for interpretation as touch, pain, or temperature. |

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
