# Peer review of "Tongue Posture, Tongue Movements, Swallowing, and Cerebral Areas Activation: A Functional Magnetic Resonance Imaging Study"

_applsci, doi:10.3390/app10176027_

Round 1

Reviewer 1 Report

This is an interesting paper addressing a very specific but complex movement condition:

Several questions are open and should be discussed:

1) what is our current basic understanding about tongue movements and how do these findings fit with standard concepts ?

2) how were movements controlled in the MR scanner ?

3) why were only young volunteers investigated while the discussion focusses on elderly people with multiple diseases ? which diseases did the authors have in mind ?

4) Why don't  we know much  about functional tongue disturbances in patients with bilateral strokes or other bilateral hemispheric lesions   ?:

5) bearing the widely distributed activity patterns in mind, how can you exclude other cortical activity patterns associated with the difficulties of the exam procedure ? Did you try non-stimulation measures ?

Author Response

) How detailed should the explanation be? What standard concept are we talking about? We addressed this issue in the updated/corrected version of the article.

2) The movements of volunteers while within the Siemens 1.5 Tesla scanner were controlled by standard built-in contraptions.

3) We shortened and edited the Discussion section and we kept only relevant references to the purpose of the study. But I'm unsure where we mentioned elderly with multiple diseases.

4) Patients with functional tongue disturbances in bilateral strokes or other bilateral hemispheric lesions were not the focus of this study. I'm unsure where we mentioned them in the study.

5) We did not exclude other cortical or subcortical activity. It's possible that in the first version of the article this relationship was not clear enough. In the last revised version, we mentioned in a few instances that the volitional nature of the test, coupled with its built-in learning task, coupled with an auditory stimulus to activate the movement, coupled with the need to accumulate saliva resulted in greater activation of cortical and subcortical areas. However, in comparing the various areas some patterns emerged that are consistent with previous studies.

Reviewer 2 Report

In this work, Scoppa and colleagues investigated cortical activations during two swallowing tasks using fMRI. It is a relatively interesting study idea however, with some experiment design and data analysis flaws. In addition to my major concerns, there are some minor issues as well.

Major concerns:

  1. Intro: The logic flow of the Introduction needs to be improved. In its current form, it is not well stated why the proposed study is significant and what gaps will it cover in the study field.
  2. Method: fMIR parameters: the authors used a thickness of 4mm in the fMRI which is relatively low for a "pinpoint" study as the authors stated. To pinpoint a cortical representation, at least a 2x2x2 resolution is recommended.
  3. Method: fMRI task: it is not clear about the total length of the tasks each participant did. it is 9m x 2 = 18m?
  4. Method: fMRI task: The boxcar design in the fMRI seems too long in each cycle. How long does an actually swallow take? I assume 2 seconds? why did the authors design a 30s duration for each cycle? This will significantly reduce the signal contrast between active and rest. Please clarify the rationale of doing so.
  5. Method: stats: in the discussion, the authors stated that there was no difference between the two tasks shown in SPM8. However, in the Methods, there is no such information provided. Please add in such between group comparison information and the corresponding hypothesis and rationale of doing so.
  6. Method: stats: in the study design, the authors stated that the participants were divided into two groups randomly with the task orders. Did the authors run a between group comparison to see if there is an order effect in each task? Please include these comparison information and the corresponding results.
  7. Method: stats: there are two significance levels used in this manuscript: 0.001 (L191) and 0.0001 (L175). Please provide rationale of doing so. Or change them to the same level and report the new results.
  8. Method: stats: L144: the authors analysed the fMRI data based on Voxel-wise analysis method, however, in further comparison, they switched to Cluster-wise analysis method. Please provide rationale of doing so. Also, how did the authors define Cluster when the fMRI data was analyzed using Voxel-wise method? Please clarify and provide a clear definition of Cluster and Cluster size.
  9. L186: It is unclear what the term "less consistent activations" means. Please clarify and rephrase.
  10. Discussion: L225-247: the discussion of others work is too much and the conclusion drawn is too spectacle without solid evidence. The authors stated later in the discussion that there was no between-task difference found in SPM8, it is not valid to discuss the potential reasons of the difference here.
  11. Discussion: the overall discussion is wordy and needs cut down. Maybe reduce the discussion of others work and focus more on the finding in the proposed study.

Minor issues:

  1. L25: functional magnetic resonance imaging -> fMRI.
  2. L25: add a space after the functional magnetic resonance imaging. and before the word "which".
  3. L43: activity -> activities.
  4. L45: remove the "and" between "advancing" and "this".
  5. L51: remove the additional space between "be" and "placed".
  6. L51: remove the additional space between "prevalent" and "activity".
  7. L56: add a space between "imaging" and "technique. " 
  8. L56: functional magnetic resonance imaging technique, or fMIR -> functional magnetic resonance imaging (fMRI) technique.
  9. L67: add reference number in the "[]".
  10. L75: becomes -> become.
  11. L84: add a space after the first sentence.
  12. L85: remove additional space between "required." and "The"
  13. L90-92: remove this paragraph which is not essential.
  14. L95: list how many channels are there in the coil used in this study.
  15. L99: please provide the resolution information of the fMRI data, 4x4x4mm^3 ?
  16. L129-135: remove the who definition text of the MNI or Tailarach. It is supposed to be known by the readers.
  17. L216-217: It has been posited as well that ....  Rephrase. It is unclear what the authors want to express in this sentence in its current form. 

Round 2

Reviewer 2 Report

Comments to the responses to the previous comments.

General comment:

Typically, when respond to reviewer’s comment, authors may want to provide the specific location of the edits caused by the corresponding responses, such as page X line XXX-XXX. Without such information, it is hard to track the responses.

There is no line numbering in the manuscript which makes it very hard to review.

Intro: The logic flow of the Introduction needs to be improved. In its current form, it is not well stated why the proposed study is significant and what gaps will it cover in the study field.

Thank you for this comment. We re-wrote the abstract and intro to reflect the purpose of this article and its clinical application.

Comment: there is still problem with the logic flow of the Introduction. It is still not clear why the comparison of the two different swallow tasks is essential. What can this kind of comparison contribute to the current research society? A typical outline may be:

  1. Intro to swallow
  2. Current swallow research progress and findings
  3. Problems with previous studies and what is needed and has not been done
  4. What study design can meet this need
  5. Current study design aims and hypothesis

Method: fMIR parameters: the authors used a thickness of 4mm in the fMRI which is relatively low for a "pinpoint" study as the authors stated. To pinpoint a cortical representation, at least a 2x2x2 resolution is recommended.

We provided a clarification in the Method section.

Comment: unfortunately, this was not addressed by the authors. Please edit and respond carefully to show respect to the reviewer’s time.

Method: fMRI task: it is not clear about the total length of the tasks each participant did. it is 9m x 2 = 18m?

We provided a clarification in that section.

Comment: reads good now.

Method: fMRI task: The boxcar design in the fMRI seems too long in each cycle. How long does an actually swallow take? I assume 2 seconds? why did the authors design a 30s duration for each cycle? This will significantly reduce the signal contrast between active and rest. Please clarify the rationale of doing so.

The rationale for this decision and further clarifications have been provided in that section.

Comment: for the statement of “a functional resonance of a just a few seconds on 4 mm thick slices would not be enough to have accurate information”, please provide reference to support this point.

For the statement of “subjects were instructed to swallow accumulated

saliva, which requires some time to naturally be produced”. How long does saliva to be accumulated enough? Can the authors provide reference to support this rationale?

Also, if the it is limited by the 4mm thickness of the slice, a reasonable solution would be using 2x2x2 mm^3 scans rather than adopting a longer duration which causes lower SNR and diminish the validity of the results.

Method: stats: in the discussion, the authors stated that there was no difference between the two tasks shown in SPM8. However, in the Methods, there is no such information provided. Please add in such between group comparison information and the corresponding hypothesis and rationale of doing so.

We corrected and re-worded that section to reflect the differences between the two tasks vis-a-vis cortical and subcortical activations.

Comments: unfortunately, no such edits in the Methods were observed in the revised manuscript.

Method: stats: in the study design, the authors stated that the participants were divided into two groups randomly with the task orders. Did the authors run a between group comparison to see if there is an order effect in each task? Please include these comparison information and the corresponding results.

No, a between group comparison was not performed, as there were only 2 groups and 2 tasks. The beginning task was randomly assigned (either protrusion or elevation) to one group and the second group was assigned the opposite order

Comment: Is the point of such randomization to avoid order effect? If not, please provide the reason why the authors randomized the two tasks order and designated them into two groups. If yes, please proof that the order effect was successfully avoided by such this randomization method. A statement like “a between group comparison was not performed”, cannot address the concern of a potential design flaw.

Method: stats: there are two significance levels used in this manuscript: 0.001 (L191) and 0.0001 (L175). Please provide rationale of doing so. Or change them to the same level and report the new results.

Yes, thank you, 0.0001 is wrong: 0.001 is correct. I corrected it

Comment: in the edited Methods, it is stated that the significance level is 0.01 (the first line on Page 5). Not 0.001 as stated in the response.  

Method: stats: L144: the authors analysed the fMRI data based on Voxel-wise analysis method, however, in further comparison, they switched to Cluster-wise analysis method. Please provide rationale of doing so. Also, how did the authors define Cluster when the fMRI data was analyzed using Voxel-wise method? Please clarify and provide a clear definition of Cluster and Cluster size.

The issue voxels-wise and cluster-wise was addressed and hopefully resolved.

Comment: It is important to keep method consistent when analyzing neuroimage. It is not common to use voxel-wise method for one purpose and cluster-wise method for another.

Also, it is still not clear how the authors define “cluster”.

Round 3

Reviewer 2 Report

  1. Introduction: the logic flow of the intro still needs improvement. Current flow: 1. what is swallow -> 2. two types of swallow -> 3. current research on the first type of swallow -> 4. importance to compare the two swallows -> 5. study aims. Maybe swap 2 and 3 to make the logic flow more straightforward.
  2. Line 135-136: the authors cited reference [77] to support the task fMRI design. The article was published in 2001 which is nearly 20 years ago. The reason that they only allowed one swallow in each task cycle is: in that study, the data processing method was to extract all epochs and average the BOLD responses in each voxel which requires that only one event can be done during each cycle. The reason that they used a 30-second duration for each task cycle is the thickness of the image they collected is 10mm which requires more time to lower the SNR. In the current study, neither the data processing method (stats test of the Beta of the GLM linear regression within each voxel) nor the thickness (4mm) is the same as the cited one. Thus, it is not appropriate to be used as a supportive reference. I suggest the authors cite more recent publications to support the study design. A study done 20 years ago diminishes the novelty and technical advantages of the recent study designs.
  3. Line 150-155: Not sure if it is appropriate to directly state "the thickness of 4mm instead of 2mm is the right compromise". At least this is still being questioned. Please cite references to support such a strong statement. Also, by reading the statement that the 4mm thickness is better than 2mm, it gives the readers an impression that the more recent/advanced studies using 2mm thickness chose an inappropriate thickness parameter. This should not be the truth.
  4. Data analysis: Did the authors use any mask? Since in the fMRI task design, the tasks was indicated by acoustic cues in each cycle, which should lead to activation in the auditory cortex. In the results, however, no such activates are shown in the figures. The participants swallowed once in each 30 seconds cycle and were acoustically cued once too. Thus it is reasonable to see activation in auditory cortex too. Please clarify this.
  5. Cross-over task study design: I understand that the cross-over design (one group did task 1 followed by task 2 and the other group did the opposite) is to cancel out potential order effect. However, the it is very important to verify that the cross-over design actually worked and did not introduce other unwanted side effects, especially when there is no wash-out period (a long gap between the two tasks, typically a day or even a week), which can lead to false positive results in the stats. For example, what if the two tasks are affected differently by the order effect? Consider the following situation: task 1 can be significantly affected by the residual effect of task 2 (which is very common in fMRI study and this is why we always want to scan resting state fMRI prior to any task fMRI) and task 2 won't be affected by task 1. In this case, for group one, task 1 and task 2 are not affected by each other, however, in group two, task 1 is significantly affected by task 2 and task 2 is not affected. When group them together without verifying the order effect, task 1 will show significant difference from task 2 by the order effect in group one. And this significant effect may not be necessarily the true difference between the two tasks. I strongly suggest the authors verify the order effect rule out such potential false significance. Otherwise, it is not appropriate to publish the results which is potentially contaminated by the side effect of back-to-back cross-over design.
  6. Previous request of "Please clarify and provide a clear definition of Cluster and Cluster size." is not acknowledged. Since the data were processed and analysed in a voxel-wise manner. It is important to clearly define how the clusters were formed by voxels for further cluster-wise analyses.
  7. Minor: Line 135-136: there are two "seconds" in this sentence.

Round 4

Reviewer 2 Report

The manuscript improved in this round.

However, the persisted denying of author to verify the order effect is disappointing.

According to the authors, in task 1 and task 2, the participants received acoustic cues. In the results of Task 2, in Figure 3 or Table 2, no auditory cortex activation was observed. In the results of Task 1, in Figure 2 and Table 1, activation in auditory cortex was presented. This is hard to explain.

Minors:

Reference numbering needs to be corrected. For example, the 11th reference cited in Introduction is numbered as [76].

Line388, there is a "(" before the reference citation.
